# Unified Multimodal Autoregressive Modeling with Shared Context —Visual Tokenizer is Key to Unification

**Wujian Peng** [1 2 * #]  **Lingchen Meng** [3 * †]  **Yuxuan Cai** [3]  **Xianwei Zhuang** [3]  **Yuhuan Yang** [3]  **Rongyao Fang** [3]
**Chenfei Wu** [3]  **Junyang Lin** [3]  **Zuxuan Wu** [1 2]  **Shuai Bai** [3]

## Abstract

Unified Multimodal Modeling aims to integrate visual understanding and generation within a single system. However, existing approaches typically rely on two disparate visual tokenizers, which splits the representation space and hinders truly unified modeling. We propose UNIAR, a unified autoregressive framework where a single discrete visual tokenizer serves as the key bridge between understanding and generation, enabling a ***shared context*** in which the model can directly interpret its own generated visual tokens without additional re-encoding. UNIAR adapts a pretrained vision encoder with multi-level feature fusion and a lookup-free bitwise quantization scheme, preserving both high-level semantics and low-level details while scaling the effective visual vocabulary at minimal cost. Building on this, the unified autoregressive model adopts ***parallel-bitwise-prediction*** to jointly predict spatially grouped, multi-level visual codes, substantially reducing visual sequence length and accelerating generation. Finally, a diffusion-based visual decoder operates on discrete visual tokens to decode high-fidelity images. Through large-scale pre-training, followed by supervised fine-tuning and reinforcement learning, UNIAR achieves state-of-the-art performance on image generation and image editing while remaining competitive on multimodal understanding benchmarks. The project homepage is at https://sharelab-sii. github.io/uniar-web.

[*]Equal contribution [#]Work done during internship at Qwen Team, Alibaba Group [†]Project Lead [1]Institute of Trustworthy Embodied AI, Fudan University, Shanghai, China [2]Shanghai Innovation Institute, Shanghai, China [3]Qwen Team, Alibaba Group, Beijing, China. Correspondence to: Zuxuan Wu <zxwu@fudan.edu.cn>, Shuai Bai <baishuai.bs@alibaba-inc.com>.

*Proceedings of the 43rd International Conference on Machine Learning*, Seoul, South Korea. PMLR 306, 2026. Copyright 2026 by the author(s).

## 1. Introduction

Recent years have witnessed significant progress in visual understanding (Li et al., 2024b; Bai et al., 2025; Peng et al., 2025) and generation (Esser et al., 2024; Wu et al., 2025a; Liu et al., 2026), largely driven by two distinct paradigms: autoregressive modeling and denoising diffusion. Increasingly, Unified Multimodal Models (UMMs) aim to integrate both capabilities into a single system (Team, 2024; Chen et al., 2025c; Deng et al., 2025; Wang et al., 2024b), enabling a model to understand and generate visual content in *shared context—i.e.* , to directly interpret its own visual generations without re-encoding. However, this integration is non-trivial due to an inherent tension in visual representation: understanding relies on high-level semantics, whereas generation demands low-level, high-frequency details. As a result, most existing works rely on *two* separate visual tokenizers (Chen et al., 2025c; Deng et al., 2025), which places understanding and generation in different representation spaces. Consequently, generated images must be re-encoded by the understanding tokenizer before the model can interpret them, breaking the desired "shared-context" goal and hindering true unification.

Using a *single* tokenizer to unify visual understanding and generation is a promising direction (Team, 2024; Wang et al., 2024b; Geng et al., 2025). However, this approach faces non-trivial challenges: (1) how to design visual tokens that simultaneously meet the needs of both generation and understanding, and (2) how to scale up the tokenizer vocabulary at minimal cost. Motivated by these challenges, we propose UNIAR, a **UNI**fied **A**uto-**R**egressive framework that leverages a *single* visual tokenizer to effectively unify generation and understanding in a shared context. To the best of our knowledge, **UNIAR** *is the first unified model to leverage multi-level bitwise visual tokens, achieving effective and efficient visual understanding and generation.*

Specifically, we fuse multi-level visual features to improve representational capacity, preserving low-level details from shallow layers and high-level semantics from deeper layers, so that the resulting representation supports both discriminative and generative tasks. Moreover, we adopt a lookup-free binary quantization scheme that maps visual features

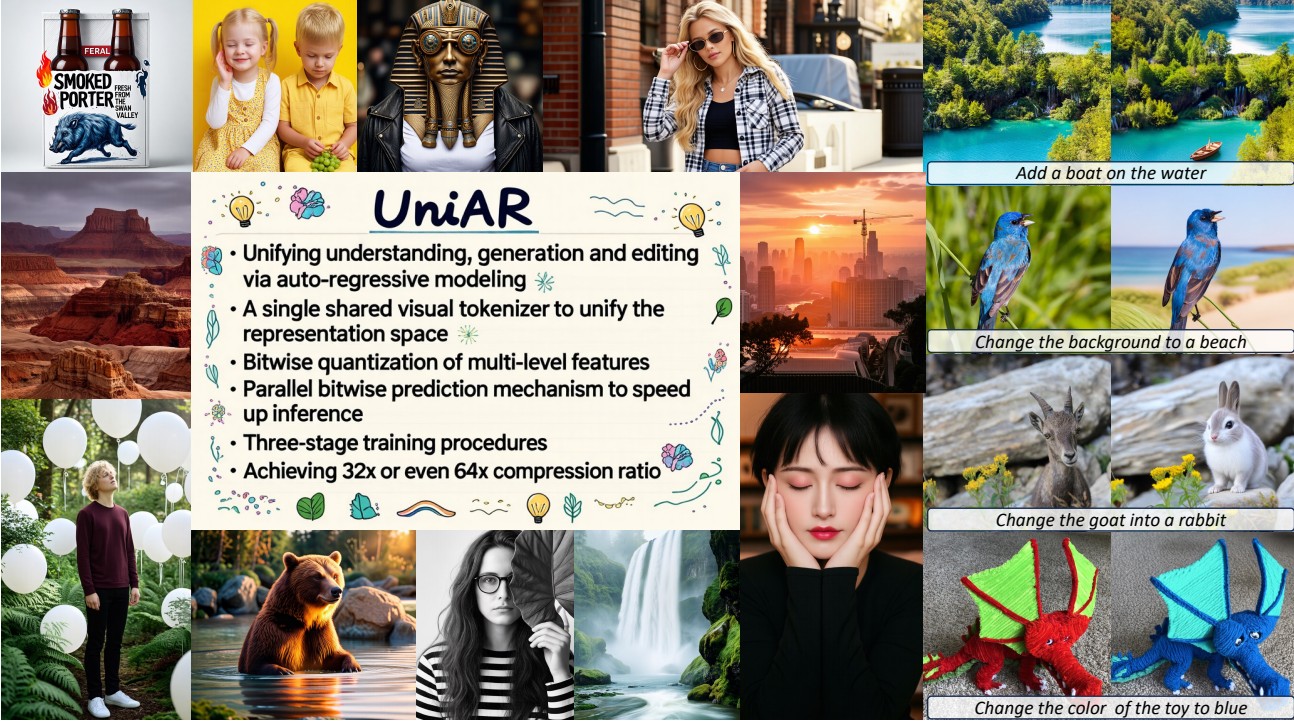

*Figure 1.* **Visual content generated by UNIAR.** UNIAR produces high-fidelity visual content with demonstrated efficacy in instruction-following and text rendering, alongside versatile image editing capabilities.

to discrete binary vectors (Zhao et al., 2025a; Han et al., 2025b). In contrast to conventional vector quantization (Van Den Oord et al., 2017), this scheme removes the need for an explicit codebook and exponentially scales the theoretical vocabulary size (*e.g.* , a 64-bit vector yields $2^{64}$ unique codes), substantially expanding the vocabulary with only a small overhead. Building on this discrete visual tokenizer, we use a unified autoregressive model under a next-token prediction paradigm to jointly model understanding, generation, and editing. We further introduce a parallel bitwise prediction mechanism that predicts multi-level visual binary vectors within each $2\times2$ spatial grid simultaneously, effectively reducing the visual sequence length and achieving $32\times$ visual compression ratio in autoregressive prediction. Finally, we employ a DiT-based visual decoder with resolution upsampling, which conditions solely on visual tokens (without text prompts) to reconstruct high-fidelity images; when employing upsampling, a $1024\times1024$ image requires predicting only 256 visual tokens.

We train the autoregressive model via three stages: large-scale pretraining on multimodal corpora, supervised fine-tuning on curated high-quality data, and reinforcement learning to further improve generation performance. Throughout the training process of the autoregressive model, the visual tokenizer and the visual decoder are kept frozen, and the decoder is introduced only during reinforcement learning to decode images for reward computation. Extensive ex-

periments show that UNIAR delivers state-of-the-art text rendering and instruction-following performance for image generation, while remaining competitive on multimodal understanding. Our contributions are three-fold:

- We propose UNIAR, a unified autoregressive framework with a *single* discrete visual tokenizer for unified modeling. In particular, multi-level feature fusion bridges the needs of generation and understanding, while lookup-free bitwise quantization scales up the tokenizer vocabulary efficiently.

- We introduce a unified autoregressive modeling paradigm with parallel bitwise prediction, together with a DiT-based decoder equipped with resolution upsampling, enabling efficient high-resolution generation with compact visual sequences.

- Extensive experiments validate the effectiveness of UNIAR, achieving state-of-the-art image generation performance while remaining competitive on standard multimodal understanding benchmarks.

## 2. Related Works

### 2.1. Visual Tokenizers

Visual understanding and generation impose distinct requirements on visual tokenization. Understanding tasks typi-

cally prioritize high-level semantic representations (Radford et al., 2021; Tschannen et al., 2025; Chen et al., 2025d), whereas generative tasks require high-frequency details such as texture and color (Van Den Oord et al., 2017; Wang et al., 2024a; Chen et al., 2026). Previous works have attempted to accommodate both objectives by integrating dual-tokenizers (Chen et al., 2025c; Tian et al., 2025b;a); however, this inherently partitions the visual context. Several studies have sought to bridge this gap by developing unified tokenizers supporting both objectives simultaneously (Lu et al., 2025; Zhao et al., 2025b; Tang et al., 2025; Han et al., 2025a) or dual-codebook design (Qu et al., 2025; Ma et al., 2025a). In this work, we employ a multi-level feature visual tokenizer that integrates features from various ViT layers to satisfy the requirements of both generation and comprehension. We quantize the visual features into discrete binary tokens, enabling autoregressive visual modeling for both tasks. In this respect, UNIAR is closely related to Infinity (Han et al., 2025b), which also adopts binary visual quantization (Zhao et al., 2025a). However, there are several differences. Infinity is designed specifically for generation, while UNIAR targets unified multimodal modeling. Moreover, Infinity is optimized for reconstruction, whereas UNIAR learns semantic multi-level features that support both generation and understanding. The two methods also differ in modeling paradigm: Infinity adopts next-scale prediction, while UNIAR uses standard next-token prediction for both vision and language.

### 2.2. Unified Multimodal Models

Existing research explores various strategies for unified modeling. One paradigm (Pan et al., 2025; Lin et al., 2025; Wu et al., 2025a) decouples tasks by interfacing a cascaded architecture composed of a frozen LMM and a diffusion model. However, these frameworks often lack architectural synergy between the understanding and generation components. Alternatively, another paradigm (Deng et al., 2025; Liu et al., 2025c; Zhou et al., 2025; Xie et al., 2025; Ma et al., 2025b) employs hybrid Transformers that generate text autoregressively and images via flow-matching. However, these approaches typically rely on divergent training objectives or disrupt the inherent causal mechanisms of standard LLMs and LMMs, which increase the training cost. Distinct from these approaches, our UNIAR achieves deep integration by employing a unified visual tokenizer to align the representation space, and modeling both generation and understanding within a unified autoregressive framework.

Among existing works, X-Omni (Geng et al., 2025) is most closely related to ours. However, **UNIAR distinguishes itself in several dimensions:** (1) Quantization: instead of an explicit codebook, we employ lookup-free bitwise quantization across multi-level features, which expands theoretical codebook size with much lower overhead; (2) Efficiency:

we introduce a parallel-bitwise-prediction paradigm, quadrupling the inference speed via reduced autoregressive steps; (3) Lightweight: UNIAR achieves competitive results with significantly fewer parameters, *i.e.* a more compact visual tokenizer (400M vs. 1B) and diffusion decoder (2.5B vs. 12B); and (4) AR-centric: X-Omni's visual decoder feeds both text prompts and visual features, while our visual decoder is independent of text inputs; it functions purely for image decoding, leaving all semantic and structural generation to the autoregressive model.

## 3. UNIAR

The overall architecture of UNIAR comprises three core components: (1) a unified visual tokenizer that encodes images into discrete semantic tokens shared across understanding, generation and editing tasks (Sec. 3.1); (2) a unified autoregressive model for the joint prediction of visual and text tokens (Sec. 3.2); and (3) a DiT-based visual decoder that translates visual tokens back into pixel space (Sec. 3.3).

### 3.1. Unified Visual Tokenizing

To facilitate autoregressive image generation, visual inputs need to be encoded into discrete tokens. Unlike traditional reconstruction-based paradigms (Zhao et al., 2025a; Van Den Oord et al., 2017; Shi et al., 2025; Yu et al., 2024), UNIAR adapts continuous features from **semantic** vision encoder into discrete representations. Specifically, we integrate Binary Spherical Quantization (BSQ) (Zhao et al., 2025a) into the visual encoder of a pre-trained Large Multimodal Model (LMM), and end-to-end finetune the model with visual understanding objectives. Unlike the conventional VQ-VAE (Van Den Oord et al., 2017), which maps visual feature to a specific index in an explicit codebook, BSQ quantizes features as a binary vector $\mathbf{u} \in \{0, 1\}^{d^{BSQ}}$. This approach enables the construction of vastly larger implicit vocabularies with minimal overhead, as the codebook size scales exponentially with the quantization dimension. In our approach, we set the BSQ dimension $d^{BSQ} = 64$, theoretically providing an expansive codebook of $2^{64}$ unique tokens. The quantization process is formally defined as:

$$
\begin{aligned}
\mathbf{v} &= \mathrm{Encoder}(x) \\
\mathbf{u} &= \mathrm{BSQ}(\mathrm{MLP}_{in}(\mathbf{v})) \\
\mathbf{v}' &= \mathrm{MLP}_{out}(\mathbf{u}) \\
\hat{\mathbf{v}} &= \mathrm{Merger}(\mathbf{v}')
\end{aligned} \quad \left.\right\} \begin{smallmatrix}\text{BSQ-added}\\ \text{parameters}\end{smallmatrix} , \tag{1}
$$

where $x$ represents the input image and $\mathbf{v}$ denotes the raw features extracted by the vision encoder. $\mathrm{MLP}_{in}$ and $\mathrm{MLP}_{out}$ refer to projection modules that map visual features to and from the BSQ quantization space, respectively. The binary vector $\mathbf{u} \in \{0, 1\}^{64}$ serves as the discrete bitwise representation, from which $\mathbf{v}'$ is reconstructed. Following

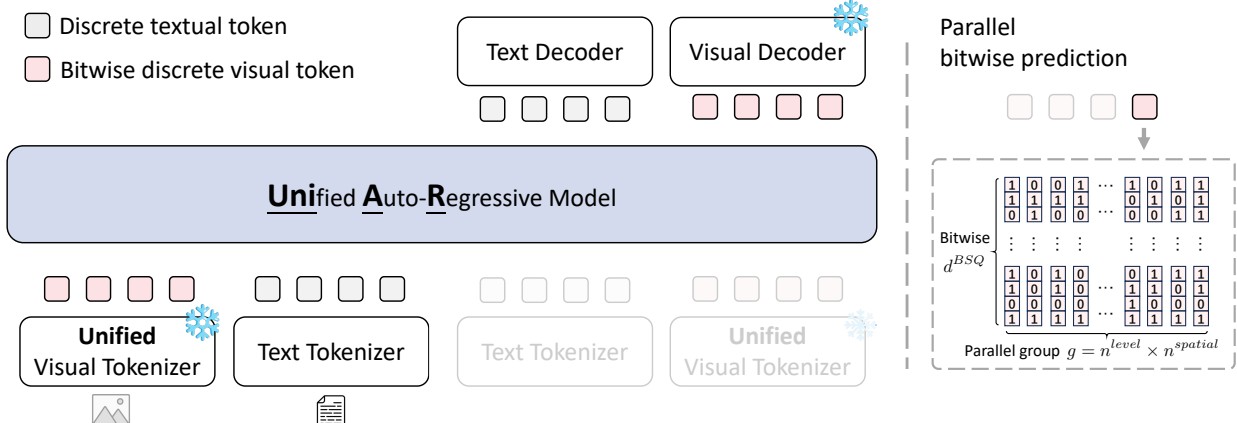

*Figure 2.* **UNIAR Framework Overview.** Our architecture integrates a **unified visual tokenizer** for bitwise quantization of image features into discrete semantic tokens, a **unified autoregressive backbone** that unifies generation and understanding via next-token prediction, and a **DiT-based decoder** for high-fidelity image decoding from predicted tokens. For visual generation, the auto-regressive model performs **parallel bitwise prediction** that predicts the next group of bit indices in parallel as shown in the right part. Notably, both the text and visual decoders are only required during inference and are not utilized during the pretraining stage.

the BSQ quantize module, a spatial merger Merger aggregates $2 \times 2$ visual features into one token $\hat{\mathbf{v}}$, projecting it to the hidden dimension of the LLM.

To further enrich the representation capacity, we adopt a multi-level fusing scheme that aggregates information across multiple layers of the visual encoder. Specifically, in addition to the final layer, we incorporate three intermediate layers into the quantization process (Meng et al., 2024; Bai et al., 2025). This hierarchical design facilitates the extraction of multi-granular visual cues for both generation and understanding tasks.

Following the original BSQ formulation, we employ the soft entropy loss for quantization. In contrast, we replace the standard Mean Squared Error (MSE) loss of reconstruction-based decoder with the Cross-Entropy (CE) loss of LMM. Formally, the overall training objective is defined as follows:

$$\mathcal{L} = \mathcal{L}^{\text{CE}} + \lambda^{\text{BSQ}} \cdot \mathcal{L}^{\text{BSQ}}, \tag{2}$$

where $\mathcal{L}^{\text{CE}}$ is the CE loss of the original LMM and $\mathcal{L}^{\text{BSQ}}$ is the original BSQ soft entropy loss. Once the discrete adaptation procedure is complete, the vision encoder is frozen to guarantee consistent visual indices for unified autoregressive modeling.

### 3.2. Unified Auto-Regressive Modeling

Our unified autoregressive model is built upon a pre-trained LLM backbone, integrated with the discrete visual tokenizer described in Sec. 3.1. We treat visual understanding and generation tasks equally, utilizing teacher-forcing paradigm for both modalities. To handle autoregressive prediction over an expansive discrete vocabulary, we implement bitwise prediction (Han et al., 2025b), applying a Cross-Entropy

loss directly to the quantized bit vectors. Furthermore, to maximize the efficiency of autoregressive modeling while minimizing the visual context length, we introduce **parallel bitwise prediction**. This strategy enables the model to concurrently predict a group of BSQ indices, leveraging the compressed representations provided by the spatial merger and multi-level DeepStack. The visual prediction head comprises an RMSNorm layer followed by a linear projection. Given the LLM hidden states $\mathbf{h} \in \mathbb{R}^{d^{\text{LLM}}}$, the visual prediction logits $\mathbf{y}$ are computed as:

$$\text{logits}^{vis} = W^{vis}(\text{RMSNorm}(\mathbf{h})), \tag{3}$$

where $W^{vis} \in \mathbb{R}^{d^{LLM} \times d^{vis}}$ is the linear layer mapping the hidden dimension to the visual prediction space. For each input visual token, UNIAR predicts the next group BSQ indices. Therefore, the output dimension is defined as $d^{vis} = 2 \times d^{BSQ} \times g$, where the group size $g = n^{level} \times n^{spatial}$, representing the number of hierarchical DeepStack levels and spatial units in the spatial merger, respectively. For text generation, we retain the original output layer of the LLM.

Finally, the total training objective, $\mathcal{L}^{AR}$ is as follows:

$$\mathcal{L}^{AR} = \mathcal{L}^{text} + \lambda^{vis} \cdot \mathcal{L}^{vis}, \tag{4}$$

where $\mathcal{L}^{text}$ and $\mathcal{L}^{vis}$ denote the auto-regressive loss of text tokens and visual tokens, respectively; $\lambda^{vis}$ is the corresponding factor of visual part. This unified scheme enables multimodal large-scale pretraining while reconciling the discrepancies between visual understanding and generation.

Furthermore, to bridge the gap between teacher-forcing training and autoregressive inference, we implement **random visual index flipping** for generation tasks, following (Han et al., 2025b). Specifically, given visual BSQ

indices $\mathbf{u} \in \{0, 1\}^{seq \times d^{BSQ}}$, we randomly invert a subset of the bits. This perturbation simulates the cumulative errors typically encountered during autoregressive sampling, while the original, uncorrupted indices are retained as ground-truth labels. This technique significantly improves the stability of the visual generation. Consequently, the model produces high-quality images even under high-temperature sampling. This flipping strategy is critical for the subsequent reinforcement learning phase, as it facilitates stable exploration in high-temperature regimes.

### 3.3. Visual Decoding

We build our visual decoder upon a pretrained single-stream Diffusion Transformer (DiT), to reconstruct pixels from the visual tokens. To integrate visual guidance, we fuse the conditioning visual signal $f_v$ with the noisy hidden states $z$ via element-wise addition (Zhang et al., 2023b; Geng et al., 2025). We optimize the decoder using Conditional Flow Matching (CFM) (Lipman et al., 2022), with training objective formulated as:

$$\mathcal{L}^{\text{CFM}} = \mathbb{E}_{t, p_t(z|\epsilon), p(\epsilon)} \|\mathcal{D}_\Theta(z \oplus f_v, t) - u_t(z|\epsilon)\|_2^2, \quad (5)$$

where $\mathcal{D}_\Theta$ denotes the visual decoder and $z \in \mathbb{R}^{h \times w \times d^{dit}}$ represents the DiT internal hidden state. To obtain the visual conditioning signal $f_v$, we process the predicted visual BSQ indices $\mathbf{u}$ as follows. At each autoregressive step, the model predicts a $2 \times 2$ spatial region, where each grid cell contains $n^{level}$ BSQ vectors from different encoder layers. For the spatial dimension, we flatten the $2 \times 2$ grid into the sequence dimension while preserving the original spatial order. For the multi-level features at each spatial position, we concatenate them along the feature dimension and project the result to $d^{dit}$, yielding $f_v \in \mathbb{R}^{h \times w \times d^{dit}}$. By fusing $f_v$ with $z$ via element-wise addition, the decoder reconstructs images with visual guidance.

To alleviate the computational burden of the autoregressive model, we introduce a resolution upsampling strategy. This enables UNIAR to synthesize high-resolution images while maintaining a low autoregressive budget. Specifically, we apply 2D bicubic interpolation to $f_v$ to reach the target resolution. Notably, we place all semantic and layout generation within the AR model, while the visual decoder functions only as a token-to-image translator. Therefore, unlike X-Omni (Geng et al., 2025) and NextFlow (Zhang et al., 2026), our decoder does **not** take any textual prompts as input, and is conditioned solely on $f_v$.

## 4. Experiments

### 4.1. Training Recipe

We employ Qwen3-8B (Yang et al., 2025) as the LLM backbone, a BSQ-quantized SigLiP2-So400M ViT (Tschannen

et al., 2025; Zhao et al., 2025a) with DeepStack (Meng et al., 2024; Bai et al., 2025) connections as the visual tokenizer, and a visual conditioned Stable Diffusion 3.5 Medium DiT (Esser et al., 2024) as the visual decoder. The training of UNIAR proceeds in a sequential, modular fashion. We first adapt the visual encoder into a discrete tokenizer via bitwise quantization (Sec. 3.1), after which it is frozen to ensure the visual codebook unchanged. We then train the DiT-based visual decoder (Sec. 3.3) to reconstruct images from the frozen encoder's discrete indices, after which it is also frozen. With both components fixed, we proceed to train the autoregressive model.

We optimize the autoregressive model using a three-stage training procedure, during which the visual tokenizer and decoder are kept frozen. Since the next-token prediction involves only discrete visual tokens, the visual decoder is omitted during the initial two stages and only introduced during reinforcement learning to decode images for reward signals acquisition. We incorporate four task-specific Transformer layers following the LLM backbone dedicated to visual generation, which mitigates task competition between generation and understanding.

**Pre-Training.** The pretraining corpus comprises approximately 1T tokens, partitioned into 800B tokens for 8K context length pretraining and 200B tokens for 32K context length pretraining. We maintain a 1:1 ratio between visual understanding and generation data. For visual generation, the maximum resolution is constrained to $512 \times 512$ pixels during the 8K stage, and subsequently increased to $960 \times 960$ pixels for the 32K stage. During the pretraining, the tokens of visual generation are formulated as:

*text prompt <image_gen> H W <vision_start> visual tokens <vision_end>,*

where the special token <image_gen> serves as a task-specific indicator for visual generation mode. To provide a spatial prior, we define $H$ and $W$ as the vertical and horizontal grid dimensions, respectively, which specify the token-count resolution for the generated output. Notably, while $\mathcal{L}^{text}$ is computed exclusively on the visual understanding datasets, $\mathcal{L}^{\text{vis}}$ is optimized across both understanding and generation tasks, enforcing a unified representation space.

**Supervised-Fine-Tuning.** For SFT, we utilize a combination of public synthetic data and re-synthesized data, incorporating prompts from (Chen et al., 2025b;a) alongside prompts sampled from our pre-training corpus. We adopt the ChatML format for visual generation. In total, the SFT stage consists of approximately 50B tokens.

**Reinforcement-Fine-Tuning.** The next-token prediction scheme in UNIAR's visual generation paradigm enables the seamless integration of reinforcement learning over discrete visual tokens. We introduce a reinforcement fine-tuning

*Table 1.* **Performance comparison on GenEval.** [†] denotes results with prompt rewriting, and [‡] denotes results from (Yan et al., 2025).

| Method | Single | Two | Counting | Colors | Position | Color Attr. | Overall |
|---|---|---|---|---|---|---|---|
| *Gen. Only Models* | | | | | | | |
| SDXL | 0.98 | 0.74 | 0.39 | 0.85 | 0.15 | 0.23 | 0.55 |
| DALLE-3 | 0.96 | 0.87 | 0.47 | 0.83 | 0.43 | 0.45 | 0.67 |
| SD3-medium | 0.99 | 0.94 | 0.72 | 0.89 | 0.33 | 0.60 | 0.74 |
| FLUX.1-dev[†] | 0.98 | 0.93 | 0.75 | 0.93 | 0.68 | 0.65 | 0.82 |
| *Unified Models* | | | | | | | |
| Emu3-Gen[†] | 0.99 | 0.81 | 0.42 | 0.80 | 0.49 | 0.45 | 0.66 |
| Show-o2 | 1.00 | 0.87 | 0.58 | 0.92 | 0.52 | 0.62 | 0.76 |
| Janus-Pro | 0.99 | 0.89 | 0.59 | 0.90 | 0.79 | 0.66 | 0.80 |
| UniWorld-V1[†] | 0.98 | 0.93 | 0.81 | 0.89 | 0.74 | 0.71 | 0.84 |
| BAGEL | 0.99 | 0.94 | 0.81 | 0.88 | 0.64 | 0.63 | 0.82 |
| BAGEL[†] | 0.98 | 0.95 | 0.84 | 0.95 | 0.78 | 0.77 | 0.88 |
| OmniGen2[†] | 0.99 | 0.96 | 0.74 | 0.98 | 0.72 | 0.75 | 0.86 |
| X-Omni[†] | 0.98 | 0.95 | 0.75 | 0.91 | 0.71 | 0.68 | 0.83 |
| Emu3.5 | - | - | - | - | - | - | 0.86 |
| NextFlow[†] | 1.00 | 0.92 | 0.75 | 0.90 | 0.76 | 0.70 | 0.84 |
| GPT-4o[‡] | 0.99 | 0.92 | 0.85 | 0.92 | 0.75 | 0.61 | 0.84 |
| UNIAR (ours) | 1.00 | 0.95 | 0.75 | 0.94 | 0.77 | 0.67 | 0.85 |
| UNIAR[†] (ours) | 0.99 | 0.96 | 0.70 | 0.93 | 0.77 | 0.83 | 0.86 |

*Table 2.* **Comparison of text rendering performance.** We conduct evaluation on the English splits of OneIG-Bench and LongText-Bench. UNIAR achieves strong text rendering performance, outperforming GPT-4o on OneIG-Bench and surpassing Gemini 2.5 Flash Image on LongText-Bench.

| Method | OneIG-EN | LongText-EN |
|---|---|---|
| *Gen. Only Models* | | |
| FLUX.1-dev | 0.523 | 0.607 |
| HiDream-I1-Full | 0.707 | 0.543 |
| Kolors 2.0 | 0.427 | 0.258 |
| Seedream 3.0 | 0.865 | 0.896 |
| Qwen-Image | 0.891 | 0.943 |
| *Unified Models* | | |
| Janus-Pro | 0.001 | 0.019 |
| BAGEL | 0.244 | 0.373 |
| OmniGen2 | 0.680 | 0.561 |
| X-Omni | 0.901 | 0.900 |
| GPT-4o | 0.857 | 0.956 |
| Gemini 2.5 Flash Img. | 0.894 | 0.869 |
| UNIAR (ours) | 0.873 | 0.917 |

stage to further refine the model's generation performance in image quality, text rendering and instruction following. The RL stage is applied exclusively to image generation task, while image editing and multimodal understanding tasks are not involved in this stage. More details about this stage can be found in Sec. A.2.

## 4.2. Main Results

To comprehensively evaluate the performance of our unified model, we conduct a systematic assessment across a suite of benchmarks spanning visual content generation, image editing, and multimodal understanding.

**Instruction Following.** GenEval (Ghosh et al., 2023) is a widely-used benchmark for evaluating instruction-following in text-to-image synthesis. As shown in Tab. 1, UNIAR achieves an overall score of 0.86, surpassing both the proprietary model GPT-4o (Hurst et al., 2024) and the generation-only model Flux.1-dev. This result demonstrates UNIAR's superior capability in precise instruction following during image generation, particularly in terms of accurate object counting, color-attribute binding, and spatial relationships.

**Text Rendering.** Text rendering is a pivotal capability for generative models, directly impacting their practical utility and user experience. However, it remains a significant challenge for existing text-to-image models. We evaluate UNIAR on the English subsets of OneIG-Bench (Chang et al., 2025) and LongText-Bench (Geng et al., 2025), with quantitative results summarized in Tab. 2. UNIAR demonstrates state-of-the-art performance in this domain. Specifically, it achieves a score of 0.873 on OneIG-EN, surpassing GPT-4o (Hurst et al., 2024). Furthermore, on LongText-EN, which is a benchmark specifically tailored for long text rendering, UNIAR reaches 0.917, outperforming Gemini 2.5 Flash Image (Comanici et al., 2025). These results

*Table 3.* **Performance comparison on ImgEdit-Bench.**

| Method | Add | Adjust | Extract | Replace | Remove | Background | Style | Hybrid | Action | Overall ↑ |
|---|---|---|---|---|---|---|---|---|---|---|
| *Gen. Only Models* | | | | | | | | | | |
| MagicBrush | 2.84 | 1.58 | 1.51 | 1.97 | 1.58 | 1.75 | 2.38 | 1.62 | 1.22 | 1.90 |
| Instruct-Pix2Pix | 2.45 | 1.83 | 1.44 | 2.01 | 1.50 | 1.44 | 3.55 | 1.20 | 1.46 | 1.88 |
| AnyEdit | 3.18 | 2.95 | 1.88 | 2.47 | 2.23 | 2.24 | 2.85 | 1.56 | 2.65 | 2.45 |
| UltraEdit | 3.44 | 2.81 | 2.13 | 2.96 | 1.45 | 2.83 | 3.76 | 1.91 | 2.98 | 2.70 |
| ICEdit | 3.58 | 3.39 | 1.73 | 3.15 | 2.93 | 3.08 | 3.84 | 2.04 | 3.68 | 3.05 |
| Step1X-Edit | 3.88 | 3.14 | 1.76 | 3.40 | 2.41 | 3.16 | 4.63 | 2.64 | 2.52 | 3.06 |
| FLUX.1 Kontext [Dev] | 4.12 | 3.80 | 2.04 | 4.22 | 3.09 | 3.97 | 4.51 | 3.35 | 4.25 | 3.71 |
| Qwen-Image-Edit | 4.38 | 4.16 | 3.43 | 4.66 | 4.14 | 4.38 | 4.81 | 3.82 | 4.69 | 4.27 |
| *Unified Models* | | | | | | | | | | |
| OmniGen | 3.47 | 3.04 | 1.71 | 2.94 | 2.43 | 3.21 | 4.19 | 2.24 | 3.38 | 2.96 |
| BAGEL | 3.56 | 3.31 | 1.70 | 3.3 | 2.62 | 3.24 | 4.49 | 2.38 | 4.17 | 3.20 |
| UniWorld-V1 | 3.82 | 3.64 | 2.27 | 3.47 | 3.24 | 2.99 | 4.21 | 2.96 | 2.74 | 3.26 |
| OmniGen2 | 3.57 | 3.06 | 1.77 | 3.74 | 3.20 | 3.57 | 4.81 | 2.52 | 4.68 | 3.44 |
| GPT-Image-1 [High] | 4.61 | 4.33 | 2.90 | 4.35 | 3.66 | 4.57 | 4.93 | 3.96 | 4.89 | 4.20 |
| UNIAR (ours) | 3.91 | 3.48 | 2.75 | 3.94 | 3.64 | 3.74 | 4.27 | 3.06 | 4.70 | 3.73 |

*Table 4.* **Comparison on multimodal understanding benchmarks.**

| Method | RLWDQA | MMMU | ChartQA | OCRBench | DocVQA | InfoVQA | MVBench |
|---|---|---|---|---|---|---|---|
| *Und. Only Models* | | | | | | | |
| LLaVA-1.6 (Liu et al., 2024b) | - | 35.1 | 54.8 | - | 74.4 | 37.1 | - |
| LLaVA-OV (Li et al., 2024b) | 66.3 | 48.8 | 80.0 | - | 87.5 | 68.8 | 56.7 |
| Qwen3-VL (Bai et al., 2025) | 71.5 | 69.6 | 89.6 | 896 | 96.1 | 83.1 | 68.7 |
| *Unified Models* | | | | | | | |
| Emu3 (Wang et al., 2024b) | - | 31.6 | - | 687 | 76.3 | 43.8 | - |
| Janus-Pro (Chen et al., 2025c) | - | 41.0 | - | - | - | - | - |
| BLIP3-o (Chen et al., 2025b) | - | 50.6 | - | - | - | - | - |
| Mogao (Liao et al., 2025) | - | 44.2 | - | - | - | - | - |
| BAGEL (Deng et al., 2025) | - | 55.3 | - | - | - | - | - |
| Show-o2 (Xie et al., 2025) | - | 48.9 | - | - | - | - | 56.4 |
| X-Omni (Geng et al., 2025) | - | - | - | 704 | 88.6 | - | - |
| UNIAR (Ours) | 68.5 | 44.3 | 75.9 | 833 | 91.4 | 70.0 | 62.3 |

underscore that UNIAR is effective in text rendering.

**Image Editing.** We evaluate the image editing capabilities on ImgEdit Bench (Ye et al., 2025), with results shown in Tab. 3. Our model achieves an overall score of 3.73, surpassing the Flux.1 Kontext (Labs et al., 2025) specifically designed for editing tasks, as well as powerful unified models such as BAGEL (Deng et al., 2025) and OmniGen2 (Wu et al., 2025b). This demonstrates that our UNIAR possesses strong image editing capabilities.

**Multimodal Understanding.** We conduct a comprehensive comparison across several multimodal understanding benchmarks, as shown in Tab. 4. UNIAR significantly outperforms existing unified models on OCR-oriented tasks, such as OCRBench (Liu et al., 2024c), DocVQA (Mathew et al., 2021), and InfoVQA (Mathew et al., 2022), as well as video understanding benchmark like MVBench (Li et al., 2024c). Our model achieves results surpassing LLaVA-Onevision on the overall performance, and even obtains performance comparable to state-of-the-art open-source VLMs, *e.g.* Qwen3-VL (Bai et al., 2025). However, we observe that performance on MMMU slightly lags behind. We attribute this gap to two primary factors: (1) Lack of pure-text data during pretraining, of which broad linguistic and factual knowledge is particularly beneficial; (2) We have not integrated reinforcement learning for understanding, which is proved to be effective for improving performance on reasoning-related tasks (Li et al., 2025; Team et al., 2025).

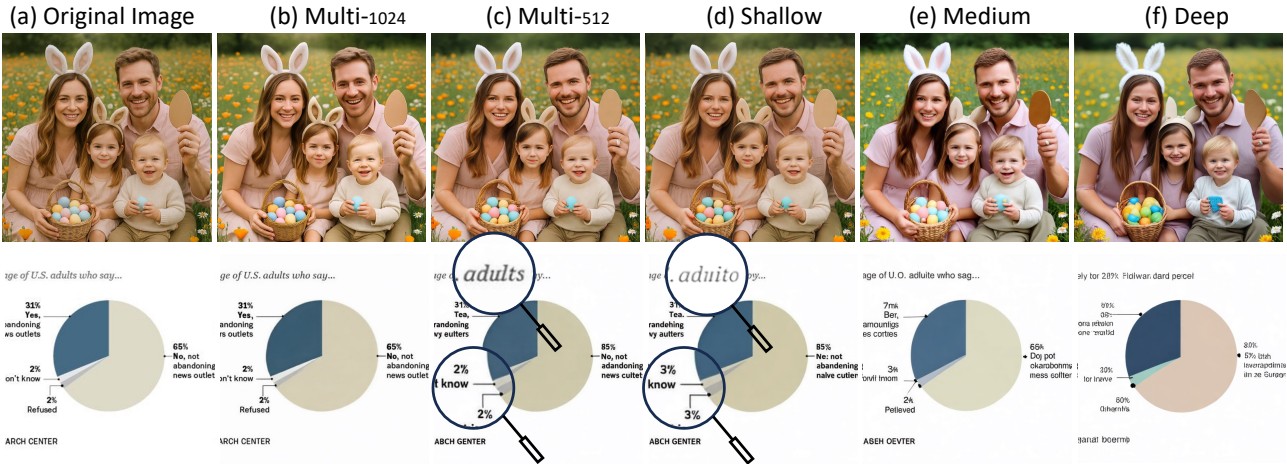

Figure 3. **Image reconstruction conditioned on different levels of visual features.** Shallow features preserve fine details while deep features capture semantics; multi-level conditioning yields the best results. Although our tokenizer is **not trained for reconstruction**, it still achieves strong reconstructions (b), highlighting the effectiveness of multi-level features.

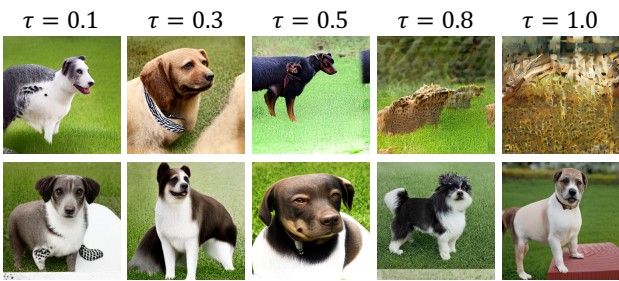

Figure 4. **Impact of random bitwise visual index flipping during pre-training.** Top: w/o flipping; Bottom: w/ flipping. Visual index flipping substantially improves generation stability.

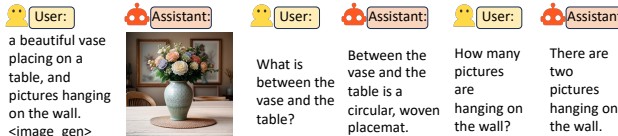

Figure 5. **Emergent interleaved generation–understanding.** Although not trained on interleaved multi-turn data, UNIAR can answer fine-grained questions about its own generated image within the same context, **without an extra visual re-encoding step**.

### 4.3. Visual Tokenizer for Multimodal Understanding

**Setup.** We follow the setup of AIMv2 (Fini et al., 2025) for multimodal instruction tuning. Specifically, we use a two-layer MLP to connect the visual tokenizer and the LLM, Llama 3.0-8B (Grattafiori et al., 2024). The parameters of the visual tokenizer are kept frozen, while the MLP connector and the LLM are trainable. We adopt the same hyperparameter settings as in AIMv2 and train the model for one epoch on the LLaVA-SFT dataset (Liu et al., 2024a).

**Evaluation.** We conduct evaluations on general knowledge tasks (OKVQA (Schwenk et al., 2022), SEED-Bench (Li

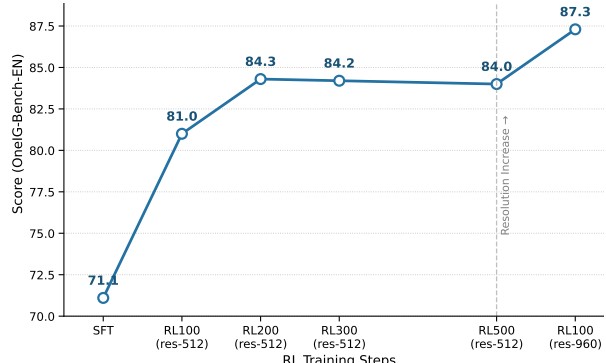

Figure 6. **Reinforcement learning significantly boosts model performance.** As the reinforcement learning steps increases, the text rendering capability improves significantly.

et al., 2024a), MME (Fu et al., 2025)) and text-rich tasks (InfoVQA (Mathew et al., 2022), TextVQA (Singh et al., 2019), DocVQA (Mathew et al., 2021), and ChartQA (Masry et al., 2022)). As shown in Tab. 5, our visual tokenizer exhibits strong multimodal understanding capabilities, achieving the best performance on TextVQA, DocVQA and ChartQA and outperforming strong baselines such as SigLIP2 (Tschannen et al., 2025) and CoMP-SigLIP (Chen et al., 2025d).

### 4.4. Ablation Studies

**Random visual index flip.** To simulate the error accumulation faced by autoregressive generation, we apply random flipping to the bitwise visual index during pretraining. As shown in Figure 4, we sample images under different temperatures and observe that the introduction of visual index flipping substantially stabilizes the generation process, enabling the model to produce coherent outputs even at higher temperatures. This improved robustness to accumulated

*Table 5.* **Results on multimodal benchmarks.** # Patches denotes the number of visual patches for the LLM.

| Model | ViT | # Patches | OKVQA | TextVQA | DocVQA | InfoVQA | ChartQA | SEED | MME |
|---|---|---|---|---|---|---|---|---|---|
| DINOv2 | L/14 | 576 | 54.1 | 13.4 | 7.3 | 21.3 | 10.8 | 57.0 | 1345 |
| CLIP | L/14 | 576 | 60.0 | 47.5 | 25.6 | 21.8 | 19.2 | 70.1 | 1481 |
| SigLIP | L/14 | 576 | 59.3 | 44.1 | 16.9 | 20.7 | 14.4 | 66.8 | 1416 |
| SigLIP 2 (NaFlex) | So/16 | 576 | 60.6 | 59.9 | 28.9 | 25.0 | 18.4 | 73.1 | 1536 |
| COMP-SigLIP | So/14 | 576 | 61.0 | 62.5 | 34.0 | 26.0 | 25.0 | 74.3 | 1543 |
| AIMv2 (336px) | L/14 | 576 | 60.8 | 53.6 | 26.6 | 22.8 | 19.2 | 71.8 | 1472 |
| UNIAR-SigLIP2 | So/16 | 576 | 60.4 | 63.1 | 38.0 | 23.8 | 26.8 | 72.8 | 1537 |

errors is particularly important for reinforcement learning, where high-temperature is needed to encourage exploration.

**Multi-level visual features.** A key design of UNIAR lies in modeling the visual tokenizer with multi-level features, which aims to preserve both low-level details and high-level semantics. To validate this, we train the visual decoder conditioned on features from different ViT layers. As shown in Figure 3, deep-layer features mainly capture high-level semantics (*e.g.* , object categories) but lose high-frequency details, whereas shallow-layer features better support fine-grained reconstruction. Conditioning on multi-level features yields the best reconstruction quality. Interestingly, although our visual tokenizer is trained for multimodal understanding rather than image reconstruction, it can still recover most visual details under the 1024-resolution setting, including accurate rendering of textual content. This unexpectedly strong reconstruction performance further confirms the effectiveness of multi-level feature modeling in UNIAR.

### 4.5. Emerging Properties.

By using a single visual tokenizer for both understanding and generation, UNIAR unifies the representation space and enables an emergent capability for interleaved generation–understanding in a shared context. Notably, our training data does **not** contain multi-turn, interleaved generation-then-understanding dialogues. Nevertheless, as shown in Figure 5, when prompted with a vague instruction, UNIAR first generates an image and can then accurately answer follow-up questions about fine-grained details that were not specified in the original prompt. This suggests that UNIAR can directly interpret its own generated visual content within the same context, without an additional visual encoding pass. In contrast, approaches that rely on separate tokenizers (*e.g.* BAGEL and Janus-Pro) require an extra re-encoding step for understanding the generated image.

### 4.6. Effectiveness of Reinforcement Learning

UNIAR utilizes a discrete visual tokenizer and performs image prediction via auto-regression. This design is highly

conducive to enhancing model capabilities through reinforcement learning. As illustrated in Figure 6, the model's text rendering performance improves significantly as the RL steps increase. When trained for 500 steps at a 512-pixel resolution, the score on OneIG-EN rose from 71.1 to 84.0. After an additional 100 steps of training at a higher resolution of 960, the metric further advanced to 87.3. This demonstrates that reinforcement learning can effectively improve the model performance.

## 5. Conclusion

We proposed UNIAR, a unified autoregressive framework that unifies understanding, generation, and editing into a single model. UNIAR is built upon three key designs. First, it adopts a unified visual tokenizer that maps both generative and understanding tasks into the same semantic space, further strengthening model unification. Second, we successfully incorporate bitwise visual tokens for unified modeling, which substantially enlarges the representational capacity while incurring less computational overhead; meanwhile, we model multi-level visual features to improve expressiveness. Third, we introduce a parallel bitwise prediction mechanism together with a visual decoder with resolution upsampling, which significantly accelerates autoregressive prediction. Experiments shown that UNIAR achieves strong performance across generation, understanding, and editing.

## Acknowledgments

This work is supported by the National Natural Science Foundation of China (Grant No. 62472098) and the Science and Technology Commission of Shanghai Municipality (No. 25511106100).

## Impact Statement

This paper presents work whose goal is to advance the field of Machine Learning. There are many potential societal consequences of our work, none which we feel must be specifically highlighted here.

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

# A. Implementation Details

## A.1. Pre-training and Supervised Fine-tuning

Our pre-training procedure is divided into two sequential stages with maximum sequence lengths of 8K and 32K tokens, respectively. We employ native resolution visual encoding during training, allowing for flexible data processing based on total token counts. For both understanding and generation tasks, we maintain a constant compression ratio of $32 \times 32$. Consequently, a $512 \times 512$ image is represented by 256 tokens for autoregressive modeling. For the 8K stage, generation resolution is capped at a resolution of $512 \times 512$ pixels. While at the 32K stage, the generation resolution limit is increased to $960 \times 960$ pixels. Images exceeding these constraints are down-scaled while preserving their original aspect ratio. During PT, SFT and RL stages, the vision encoder remains frozen, while the Large Language Model and the spatial merger are fully trainable. Data is structured into three distinct formats to support unified modeling:

- visual understanding: {prompt; **image_tokens**; **answers**} ;

- text-to-image generation: {prompt; **image_tokens**}

- image editing: {prompt; **reference_image_tokens**; **image_tokens**}.

The bold parts indicate the tokens that contribute to the autoregressive loss. For subsequent Supervised Fine-Tuning (SFT), we adapt these templates into the ChatML format to facilitate multi-turn dialogue and instruction following. We list the detailed training recipe in Tab. 6.

*Table 6.* **Training hyperparameter configurations of UNIAR.**

| Hyperparameters | PT-8K | PT-32K | SFT |
|---|---|---|---|
| Learning rate (max:min) | 5e-4:5e-5 | 5e-5:1e-5 | 1e-5:1e-6 |
| LR scheduler | Cosine | Cosine | Cosine |
| Weight decay | 0.05 | 0.05 | 0.05 |
| Gradient norm clip | 1.0 | 1.0 | 1.0 |
| Optimizer | AdamW ($\beta_1 = 0.9$, $\beta_2 = 0.98$) | | |
| Loss weight (Text: Vis) | 1:10 | | |
| Warm-up steps | 500 | 500 | 500 |
| Global batch size | 512 | 128 | 64 |
| Training tokens | 800B | 200B | 50B |

## A.2. Reinforcement Learning

**Reward system.** We adopt multiple rewards to improve different aspects of the model. Specifically, for image quality, we use HPSv2 (Wu et al., 2023) and UnifiedReward (Wang et al., 2025b) to enhance aesthetic quality and reduce artifacts. For text rendering, we apply PaddleOCR (Cui et al., 2025a) to recognize texts in generated images and compute the reward based on the edit distance to the ground-truth text, encouraging accurate text rendering. For instruction following, we use the object-detector-based reward from FlowGRPO (Liu et al., 2025a), which checks whether the generated image correctly reflects fine-grained concepts in the prompt, including object categories, counts, attributes, and relations. Finally, all rewards are normalized to the range $[0, 1]$ and averaged as the final reward for each sample.

**Data collection.** To maximize coverage across diverse domains, we curate prompts from multiple sources, including (1) prompts sampled from our SFT data, (2) open-source datasets such as BLIP3o-60k (Chen et al., 2025b) and ShareGPT-4o-Image (Chen et al., 2025a), the GenEval training set and the OCR training set created by FlowGRPO (Liu et al., 2025a), and (3) long-text prompts synthesized using a large language model.

**Training details.** We optimize the model with GRPO (Shao et al., 2024), using a constant learning rate of $5 \times 10^{-6}$. To prevent over-optimization, we add a KL-divergence term to the training loss with a coefficient of 0.01. Each batch contains 32 randomly sampled prompts, and we generate 16 images for each prompt. We train in two stages: (1) 500 steps at $512 \times 512$ resolution to quickly improve image quality and instruction following; (2) 100 additional steps at a higher $960 \times 960$ resolution to enhance long-text rendering.

*Table 7.* Training efficiency comparison between continuous and discrete visual tokens.

|  | Elapsed time per iteration (s) |
| --- | --- |
| Continuous visual tokens | 35.4 |
| Discrete visual tokens | 24.5 |

*Table 8.* GPU hours for each training stage.

|  | PT-8K | PT-32K | SFT | RL |
| --- | --- | --- | --- | --- |
| GPU Hours | 19k | 10k | 2k | 1.9k |

*Table 9.* Inference efficiency comparison.

|  | Downsample ratio | #Token | Time (s) |
| --- | --- | --- | --- |
| Janus-Pro (7B) | $\times 16$ | 4096 | 101.9 |
| X-Omni (7B) | $\times 16$ | 4096 | 119.7 |
| UniAR (8B) w/o decoder upsample | $\times 32$ | 1024 | 53.5 |
| UniAR (8B) w/ decoder upsample | $\times 64$ | 256 | 13.0 |

### A.3. Evaluation Configurations

**Autoregressive sampling.** For both text-to-image generation and image editing, we adopt classifier-free guidance (CFG) (Ho & Salimans, 2022) during *autoregressive* sampling to improve instruction following. Specifically, we compute the final prediction by combining the conditional and unconditional predictions with a guidance scale $s = 2.5$ for GenEval and $s = 2.0$ for other benchmarks. During evaluation, we set the sampling temperature to 0.1.

**Inference resolutions.** On GenEval and ImgEdit, we autoregressively generate $512 \times 512$ visual tokens and apply $\times 2$ upsampling in the DiT decoder to obtain $1024 \times 1024$ output images. For text-rendering benchmarks (LongText Bench and OneIG-Bench), we autoregressively generate $1280 \times 704$ images and do not use decoder super-resolution.

### A.4. Training and Inference Cost Analysis

Regarding training efficiency, discrete visual tokens improve pre-training throughput by approximately 30% over continuous tokens at a sequence length of 8K, as visual inputs can be pre-tokenized and stored as bit-packed representations offline. Detailed training elapsed times are provided in Tab. 7. We also report the training time for each stage in Tab. 8.

Regarding inference efficiency for image generation, we compare UniAR with Janus-Pro (Chen et al., 2025c) and X-Omni (Geng et al., 2025). We report the time required for the autoregressive generation stage, excluding the cost of the visual decoder. All models are evaluated on the same A100 GPU without CFG, and we report the time required to generate a 1024-resolution image. UniAR achieves significantly faster generation, mainly due to its higher $32\times$ downsampling ratio, which quadratically reduces the number of prediction steps. When the decoder uses upsampling, the number of predicted tokens is further reduced to 256, leading to an additional speedup.

## B. Limitations and Future Works

Due to resource constraints, we do not include pure-text data in joint pre-training, leaving considerable room for optimizing the data mixture for both understanding and generation. In addition, our RL training is currently limited to image generation. In the future, we aim to further explore large-scale pre-training by scaling both the dataset size and model parameters. Furthermore, we find that there is much room in our post-training, particularly through Reinforcement Learning. We will focus on developing reward models for specific domains, such as aesthetics, instruction-following, and text-rendering. Finally, we plan to extend the capabilities of each individual domain while exploring the synergy between them.

