# OpenReview forum: "Unified Multimodal Autoregressive Modeling with Shared Context—Visual Tokenizer is Key to Unification"
_ICML.cc/2026/Conference — ICML 2026 regular_

### Official Review · Reviewer_VbQW · 2026-02-22

**Soundness:** 3
**Presentation:** 3
**Significance:** 3
**Originality:** 3
**Overall Recommendation:** 4
**Confidence:** 4

**Summary:**

This paper presents an alternative and novel approach for establishing unified multimodal models. Different from current state-of-the-art decoupled paradigms, this paper proposed to train a unified visual tokenizer using BSQ and instantiating from understanding oriented encoders. Building on this conceptual framework, the authors conducted extensive training (pretrain, sft and rl) and evaluated on comprehensive benchmarks, indicating that the proposed shared tokenizer and UMM setup is a competitive alternative approach for unified modeling.

**Compliance With Llm Reviewing Policy:**

Affirmed.

**Final Justification:**

In this paper, the authors presented a well-designed approach for designing tokenizers for unified modeling. Through a series of well-crafted designs, the authors enabled understanding-trained tokenizers to be capable of both understanding and strong generation related capabilities. I appreciate the sound rebuttal provided by the authors and is overall positive about the technical soundness of the paper, therefor I am raising my score to weak accept.

**Key Questions For Authors:**

1. The visual tokenizer design seems a little confusing. How was the hierarchical and multi-granularity tokenization implemented, and does it incur/necessitate an enlarged token sequence count? If so, how was it handled?
2. How was the unified tokenizer trained? The authors only mentioned training with understanding objectives in Sec 3.1, it would be best to elaborate on the training data scale, specific tasks and formulation. Additionally, how did understanding oriented training ensure genuine generative reconstruction quality that reserve fine-grained information, especially for highly detailed visual text rendering tasks?

**Limitations:**

Yes.

**Strengths And Weaknesses:**

Strengths:
1. This paper presents a novel attempt for unified multimodal modeling with a single shared visual tokenizer. By designing an understanding driven BSQ tokenizer, this paper presents an alternative solution to contemporary decoupled representation UMMs.
2. Extensive and comprehensive evaluations. The author conducted thorough evaluations for the proposed UMMs on image generation, editing and various understanding benchmarks, validating the effectiveness of the proposed approach.

Weaknesses:
1. Certain key details involving the architecture and training formulation of the key unified visual tokenizer seems unclear. See questions section for detailed questions that need more elaboration.
2. Missing statistics for visual tokenizer: Visual tokenizer is a key contribution of this paper. But the authors did not report key performance statistics regarding the tokenizer. The authors should present reconstruction performance (key to generation) including rFID, PSNR and SSIM, as well as understanding performance (as the tokenizer seems to be trained on understanding oriented objectives).

---

> ### Author Rebuttal · Authors · 2026-03-31
>
> Dear reviewer VbQW,
>
> We sincerely appreciate your valuable feedback and insightful comments, which are greatly helpful in improving our work. Below, we provide detailed responses to each weakness (W) and question (Q).
> ***
> > **[W1, Q1] Clarification of the visual tokenizer architecture design**
>
> In addition to the final layer, we also introduce features from three intermediate ViT layers, and the features from each layer are quantized using BSQ. This **does not increase the sequence length**:
>
> * At the input side, these features are not concatenated along the sequence dimension. Instead, they are injected into different layers of the LLM, which preserves multi-level feature information without increasing the sequence length.
> * At the output side, features from multiple levels are predicted in parallel through the bitwise parallel prediction mechanism, and therefore do not increase the number of autoregressive decoding steps.
>
> Thank you for raising this question. We will provide more details in the revised version.
>
> ***
> > **[Q2] Clarification of the visual tokenizer training formulation**
>
> The unified tokenizer is trained on pure vision-language understanding data using the BSQ module. Specifically, we perform continuous pre-training based on a 30B MoE VL model utilizing a 3-level DeepStack connection [1, 2]. The BSQ module is inserted before the spatial merger of the last ViT layer and the DeepStack layers.
>
> We employ a combined loss function consisting of the large language model (LLM) cross-entropy loss and the BSQ loss [3], as defined in Eq. 2 of our original paper:
>
> $\mathcal{L} = \mathcal{L}^{CE} + \lambda^{BSQ} \cdot \mathcal{L}^{BSQ}$
>
> where $\mathcal{L}^{CE}$ represents the next-token prediction loss for the input images and queries, and $\mathcal{L}^{BSQ} = \mathbb{E}_u [H(\hat{q}(c|u))] - \gamma H(\tilde{q}(c))$.
>
> We train on ~150B tokens for the BSQ adaptation. Thanks to the multi-layer DeepStack, we find that the quantized features across multiple layers have sufficient potential to preserve rich information for reconstruction, enabling high-fidelity text rendering without requiring additional training heuristics.
>
> ***
> > **[W2] Statistics for the visual tokenizer**
>
> Thanks for the suggestion.
>
> `Reconstruction` We compare UniAR with several recent unified tokenizers on ImageNet val set. UniAR is evaluated at its native 1024 resolution, while the baselines use their native 256 setups; thus, this is not a strictly apples-to-apples comparison, but a practical reference under each method's training setting. Even **without any explicit reconstruction loss**, UniAR shows strong reconstruction capability, and scaling further improves performance: UniAR* scales the decoder to 8B, while UniAR** further increases the encoder BSQ dim to 128.
>
> ||rFID↓|PSNR↑|SSIM↑|
> |-|-|-|-|
> |Show-o|3.5|21.3|0.59|
> |TokenFlow|1.37|21.4|0.69|
> |VQRAE|1.3|22.2|0.76|
> |UniAR|1.4|22.7|0.71|
> |UniAR*|0.95|22.7|0.73|
> |UniAR**| 0.74|25.8|0.81|
>
> (The baseline results are taken from [4].)
>
> `Understanding` We follow the setting and hyperparameters of AIMv2 [5], freeze the visual tokenizer, and connect it to LLaMA 3.0 8B via a two-layer MLP. The model is trained on LLaVA-SFT-665k for one epoch. We fix the number of visual patches to 576 for a fair comparison. As shown below, despite being a quantized discrete tokenizer, UniAR still achieves strong understanding performance.
> |Model|#patches|TextVQA|DocVQA|ChartQA|SEED|MME|
> |---|---:|---:|---:|---:|---:|---:|
> |CLIP L/14|576|47.5|25.6|18.5|70.1|1481|
> |SigLIP L/14|576|44.1|16.9|14.4|66.8|1416|
> |SigLIP2 So/16|576|59.9|28.9|18.4|73.1|1536|
> |AIMv2 L/14|576|53.6|26.6|19.2|71.8|1472|
> |UniAR-ViT|576|59.7|30.7|19.9|71.5|1448|
>
> ***
> [1] Qwen3-VL Technical Report
>
> [2] DeepStack: Deeply Stacking Visual Tokens is Surprisingly Simple and Effective for LMMs
>
> [3] Image and Video Tokenization with Binary Spherical Quantization
>
> [4] VQRAE: Representation Quantization Autoencoders for Multimodal Understanding, Generation and Reconstruction
>
> [5] Multimodal Autoregressive Pre-training of Large Vision Encoders

---

> > ### Author Rebuttal · Reviewer_VbQW · 2026-04-02
> >
> > Thanks to the authors for the thorough rebuttal and my concerns have been partially addressed. I would like the authors to further clarify on the following:
> >
> > 1. Is there a trade off between understanding and generation (reconstruction) in UniAR? Specifically, I would like to see results on how understanding/generation results evolve for the visual tokenizer, this will help readers gain a better understanding on the mechanisms of how understanding/generation trade-off/co-improve in the proposed setting. It would also be beneficial if the authors could try different settings and report the curve between understanding and generation related metrics, to investigate whether mutual synergy exists.
> >
> > 2. The comparisons in W2 seems a little misleading. The authors should (i) make a fair comparison, keeping the resolution fixed at either 256 or 1024 for all reported models. (ii) referencing performance for some generalist specialist tokenizers under the evaluated setting, I understand that UniAR is potentially not able to match the performance of these specialist tokenizers, but a reference for how large the gap is will be beneficial.
> >
> > Overall, I appreciate the thorough rebuttal by the authors. If my final concerns can be resolved, I will be willing to raise my score.

---

> > > ### Author Response · Authors · 2026-04-05
> > >
> > > Dear Reviewer,
> > >
> > > We sincerely thank you for your thoughtful feedback and are happy to provide further clarifications.
> > > ***
> > > > **[Q1] Evolution of generation and understanding**
> > >
> > > To balance the features required for different task, we adopt a hierarchical multi-granularity feature fusion design that integrates shallow and deep-layer features, preserving the low-level details needed for reconstruction and the high-level semantic information required for understanding.
> > >
> > > To study how understanding and generation evolve, we evaluate tokenizers at different stages of tokenizer training, namely 5k, 10k, and 20k steps. We emphasize that these numbers refer to tokenizer checkpoints, rather than to the training steps used in the downstream understanding or reconstruction evaluations.
> > > * For understanding evaluation, we follow the AIMv2 [1] by freezing the visual tokenizer, connecting it to LLaMA 3 8B via a two-layer MLP, and training on LLaVA-SFT-665k.
> > > * For reconstruction evaluation, we freeze the visual tokenizer, attach a DiT-based decoder, and train the decoder for reconstruction using the setup described in Sec. 3.3 of the main paper.
> > >
> > > **Due to time and computational resource constraints, we are unable to run to full convergence**. Therefore, we present an early-stage analysis based on loss curves, using the training loss for understanding and the flow matching loss for reconstruction as proxies for their respective capabilities:
> > > * [Figure 1, understanding loss curve](https://anonymous.4open.science/r/rebuttal-figures-E33D/understanding_loss.png): The three curves correspond to tokenizers trained for 5k, 10k, and 20k steps, respectively. We observe a consistent decrease in understanding loss from the 5k tokenizer to the 20k tokenizer, suggesting that the understanding capability improves as tokenizer training progresses.
> > > * [Figure 2, reconstruction loss curve](https://anonymous.4open.science/r/rebuttal-figures-E33D/reconstruction_loss.png): The three curves correspond to tokenizers trained for 5k, 10k, and 20k steps, respectively. We likewise observe a consistent decrease in reconstruction loss as tokenizer training progresses, indicating that the reconstruction capability is also steadily improved.
> > > * [Figure 3, reconstruction visualization](https://anonymous.4open.science/r/rebuttal-figures-E33D/reconstruction_visulization.jpg): In addition, we visualize the reconstructed images corresponding to tokenizers at different training stages. Due to limited training resources, **these visualizations are obtained from an early stage of decoder training**. For a fair comparison, all decoders are trained for the same number of steps, and the only varying factor is the training stage of the tokenizer. Therefore, the reconstruction differences mainly reflect the reconstruction capability of the tokenizer itself. We can observe that the reconstructions improve as tokenizer training progresses, which is consistent with the reconstruction loss curves.
> > >
> > > These results suggest that **understanding and generation improve simultaneously during tokenizer training** (from 5k to 20k steps), indicating that UniAR achieves a favorable balance between the two rather than showing an obvious trade-off. Although these results are preliminary, they already show a clear trend, and we will provide a more complete analysis in the revised version.
> > > ***
> > > > **[Q2] Reconstruction performance comparisons at 1024 resolution**
> > >
> > > We re-evaluated Show-o and TokenFlow at 1024 resolution. We could not evaluate VQRAE because its pretrained weights are not publicly available. As suggested, we include several reconstruction-specialized tokenizers, including VQGAN, LlamaGen, and VAR, to provide a clearer reference for the gap between UniAR and specialist tokenizers.
> > > Notably, UniAR is a semantic tokenizer trained **without any reconstruction loss**, yet it still delivers competitive reconstruction performance. This finding is in line with the observation in [2] that semantic tokenizers can still achieve strong reconstruction performance.
> > >
> > > (All results are presented at 1024 resolution for a fairer comparison)
> > > ||reconstruction-loss|rFID↓|PSNR↑|SSIM↑|
> > > |-|:-:|-|-|-|
> > > |_**Reconstruction Specialist Tokenizer**_|||||
> > > |VQGAN|$\checkmark$|1.02|25.7|0.72|
> > > |LLamaGen|$\checkmark$|**0.64**|26.4|0.77|
> > > |VAR|$\checkmark$|1.91|25.6|**0.82**|
> > > |_**Unified Generalist Tokenizer**_|||||
> > > |Show-o|$\checkmark$|1.10|**26.5**|0.77|
> > > |TokenFlow|$\checkmark$|1.33|26.3|0.76|
> > > |UniAR|$\times$|1.40|22.7|0.71|
> > > |UniAR*|$\times$|0.95|22.7|0.73|
> > > |UniAR**|$\times$| 0.74|25.8|0.81|
> > >
> > > ***
> > > We again thank you for your thoughtful feedback and suggestions. We will incorporate these clarifications in the revised version and provide more detailed analysis of the tokenizer. We hope our response can addresses your final concerns.
> > >
> > > Best regards,
> > > The authors
> > >
> > > ***
> > > [1] Multimodal Autoregressive Pre-training of Large Vision Encoders
> > > [2] Diffusion Transformers with Representation Autoencoders

---

### Official Review · Reviewer_GJ3m · 2026-03-07

**Soundness:** 3
**Presentation:** 3
**Significance:** 3
**Originality:** 3
**Overall Recommendation:** 4
**Confidence:** 4

**Summary:**

This paper proposes a unified AR model with a single, multi-level bitwise visual tokenizer, that unifies the visual generation and understanding tasks. After going through the pretraining, SFT, and RL stages, the model achieves SOTA performance on some image generation tasks while maintaining competitive performance on multimodal understanding tasks.

**Compliance With Llm Reviewing Policy:**

Affirmed.

**Final Justification:**

The rebuttal has addressed all my concerns. Thus, I increase the confidence score for my positive rating.

**Key Questions For Authors:**

1. I’m still a little confused by the training protocol.

    In Line 614:
    > “During PT and SFT, the vision encoder remains frozen, while the Large Language Model and the spatial merger are fully trainable. ”

    While in Line 195:
    >“Once the discrete adaptation procedure is complete, the vision encoder is frozen and to guarantee visual indices consistent for the unified auto-regressive modeling.”

- Is the visual encoder always frozen before the RL phase, or is the visual encoder trainable in _Section 3.1_ and then kept frozen till the RL phase?
- Could the authors further explain why, during the RL phase, the visual decoder is no longer frozen as in PT and SFT?
- Also, when did DiT in _Section 3.3 Visual Decoding_ get trained?

2. Could the authors further clarify how $f_v$ is constructed in Line 234? How are different groups/levels of BSQ indices assembled and input into the MLP layers to obtain $f_v$?

3. Could the author explain why "the RL stage is applied exclusively to image generation task, while image editing and multimodal understanding tasks are not involved in this stage" (Line 251)?

4. Could the author further clarify the innovations of UniAR compared to Infinity [1]? Does UniAR follow a similar training framework to Infinity in image generation tasks?

Reference:
[1] Han, Jian, et al. "Infinity: Scaling bitwise autoregressive modeling for high-resolution image synthesis." _Proceedings of the Computer Vision and Pattern Recognition Conference_. 2025.

**Limitations:**

Yes.

**Strengths And Weaknesses:**

Strengths:
- UniAR demonstrates the capability of the binary-quantized discrete image tokenizer to unify visual generation and understanding in one AR model. This would be of great benefit for scaling up the visual vocabulary, reducing model sizes and increasing inference efficiency.

Weaknesses:
- For multimodal understanding benchmarking, in comparisons where UniAR performs strongly, only a few unified models are evaluated on the same dataset. In addition, most benchmark datasets on which UniAR achieves comparable performance focus on document understanding task (OCRBench, DocVQA, InfoVQA, ChartQA).

---

> ### Author Rebuttal · Authors · 2026-03-31
>
> Dear Reviewer GJ3m,
>
> We sincerely thank you for your time and thoughtful, constructive feedback. Below, we provide detailed responses to each weakness (W) and question (Q).
> ***
> > **[W1] Multimodal understanding benchmarks and evaluate coverage**
>
> We have further evaluated our model on additional real-world and video benchmarks. As shown in the table, our model achieves comparable and even superior performance compared with LLaVA1.6-7B (a strong open-source academic baseline) and Qwen3VL-8B-Thinking (one of the strongest open-source models).
> ||realworldqa|ai2d|mvbench|videomme_wosub|CharadesSTA|
> |-|-|-|-|-|-|
> |Llava1.6-7b|66.3|81.4|56.7|58.2|–|
> |Qwen3VL-8B-thinking|73.5|84.9|68.7|71.8|59.9|
> |UniAR|68.5|76.2|62.3|59.2|56.9|
>
> It is worth noting that we do not incorporate pure-text data during pre-training, nor do we apply RL for multimodal understanding. This demonstrates that our model has significant potential for further improvement for multimodal understanding tasks with a more balanced data mixture, higher-quality data, and multimodal RL.
> ***
> > **[Q1] Clarification of the training protocol across different stages**
>
> We clarify that the visual encoder and decoder are both frozen during the PT, SFT,_**and RL**_ phases. Our training protocol is as follows:
> * `Stage 1` Discrete Adaptation: The visual encoder is trainable only in this stage to adapt it into a binary discrete encoder via bitwise quantization. Once completed, it is frozen for all subsequent stages, including decoder training, PT, SFT, **and RL**.
> * `Stage 2` Visual Decoder Training: After the encoder is fixed, we train the DiT-based visual decoder to reconstruct images from the indices produced by the frozen encoder. After this, the decoder is also frozen for all subsequent stages.
> * `Stage 3` Autoregressive Modeling: During PT, SFT, and RL, both the visual encoder and decoder remain frozen. Only the LLM backbone and the spatial merger are trained. The decoder is not needed in PT/SFT (which predicts indices); in the RL phase, the frozen decoder is used to decode the generated visual indices into images for reward evaluation.
>
> This design of frozen encoder and decode allows us to tokenize images into discrete indices offline, which removes the need for the visual encoder during LLM training, thereby substantially improving training efficiency.
>
> We will adjust the order of Sec. 3.2 and Sec. 3.3 to clarify the training pipeline and avoid potential confusion in the revised version.
> ***
> > **[Q2] Construction of $f_v$ representations**
>
> At each autoregressive step, the model predicts a spatial region of size $n^{spatial}\times n^{spatial}$. Each grid contains $n^{level}$ features from different levels. For the spatial dimension, we directly flatten it into the sequence dimension while preserving the original spatial order. For the multi-level features, we concatenate them along the feature dimension and then use a linear layer to project them to the dimension required by the DiT. We will further clarify these details in the revised version.
>
>
> ***
> > **[Q3] RL-stage design choices**
>
> There have already been several explorations of applying RL to multimodal understanding, whereas studies on image generation, especially autoregressive image generation, are still under-explored. Therefore, we mainly aims to explore the effect of RL on autoregressive generation with discrete tokenizers. We do not include image editing as robust reward models are still limited. We find in our experiments (table below) that applying RL only to the generation task not only preserving its original understanding performance but also improving its image editing ability.
> ||DocVQA|InfoVQA|RLWDQA|ImgEdit|
> |-|-|-|-|-|
> |Before RL|69.6|835|69.3|3.58|
> |After RL|70.0|833|68.5|3.73|
>
> UniAR supports unified RL across multiple tasks, and we believe extending RL to these tasks could further improve understanding and editing performance. We will explore this in our future work.
>
> ***
> > **[Q4] Differences from Infinity**
>
> Infinity is an outstanding work, and we have discussed it. UniAR and Infinity differ in:
> 1. `Core Objective: Unification vs. Gen-only.`
> Infinity focuses solely on image synthesis, whereas UniAR is designed for unified multimodal modeling with a single visual tokenizer for both understanding, generation and editing.
> 2. `Reconstruction v.s. Semantic Encoder.` Infinity is optimized for reconstruction, with an emphasis on capturing low-level details. UniAR is trained to learn semantic multi-level features that support both generation and understanding.
> 3. `Modeling Paradigm: AR vs. VAR.`
> Infinity uses VAR with next-scale prediction, while UniAR uses standard AR with next-token prediction for both vision and language.
> 4. `Multi-level Feature Fusion.`
> UniAR fuses multi-layer features: shallow layers support generation, and deep layers support understanding.
>
> Regarding training framework, both UniAR and Infinity predict binary visual indices for image synthesis.

---

> > ### Author Rebuttal · Reviewer_GJ3m · 2026-04-03
> >
> > Thanks for the response. All my concerns have been addressed.
> >
> > Given that my original score is already positive, I'll keep the original rating.

---

> > > ### Author Response · Authors · 2026-04-05
> > >
> > > Dear Reviewer,
> > >
> > > We are glad that our previous response helped address all of your concerns, and we are encouraged that you recognize our method as offering "great benefit for scaling up the visual vocabulary, reducing model sizes, and increasing inference efficiency."
> > > We sincerely thank you again for your constructive feedback and will incorporate the corresponding revisions in the revised version.
> > >
> > > Best regards,
> > > The authors

---

### Official Review · Reviewer_VQ3s · 2026-03-12

**Soundness:** 2
**Presentation:** 3
**Significance:** 3
**Originality:** 2
**Overall Recommendation:** 4
**Confidence:** 3

**Summary:**

This paper proposes UNIAR, a unified autoregressive framework that employs a single discrete visual tokenizer to jointly support image understanding and generation. The method adopts a lookup-free bitwise quantization scheme, and parallel bitwise prediction to jointly predict spatially grouped multi-level visual codes, thereby reducing the visual sequence length and accelerating generation. A diffusion-based visual decoder is trained to reconstruct images and is conditioned solely on visual signals. The proposed method is evaluated on a range of benchmarks, including instruction following, text rendering, image editing, and multimodal understanding.

**Compliance With Llm Reviewing Policy:**

Affirmed.

**Final Justification:**

Thanks to the authors for their efforts in the rebuttal. Most of my concerns have been addressed, and I have therefore raised my score to weak accept. I hope the authors can include the additional experiments and discussion in the final version.

**Key Questions For Authors:**

1. My main concern is that the proposed method exhibits only moderate performance overall. In particular, its results on text rendering, image editing, and multimodal understanding tasks lag behind many existing models. For example, the performance on MMMU appears relatively weak. Could the authors provide more discussion or analysis on the reasons for this relatively limited performance? Would further scaling of the model or training data potentially improve the results?
2. In addition, including a comparison of model parameters and possible training costs could be better, as this would allow readers to better assess the efficiency of the proposed framework and understand whether the unified design offers practical advantages.
3. Finally, more comprehensive ablation studies on key design choices mentioned above should be added, such as the BSQ dimension, group size, and the weighting factor of the visual loss. If running additional experiments is difficult, it would still be valuable for the authors to provide more detailed reasoning or empirical considerations behind the chosen hyperparameters.

I would consider raising my score if most of these concerns can be addressed.

**Limitations:**

The discussion of the limitations is missing in the paper.

**Strengths And Weaknesses:**

Strengths:
1. The paper is generally well written and easy to follow.
2. Extensive evaluations are conducted across multiple tasks, including multimodal understanding and different generation tasks.
3. The lookup-free bitwise quantization scheme is interesting, and the diffusion-based visual decoder conditioned solely on visual signals differs from many existing diffusion generation frameworks, which might inspire other works.
4. The analysis on related works is comprehensive.

Weaknesses:

1. The reported performance lags behind several existing models on multiple benchmarks. For example, the method underperforms models such as Qwen-Image and Gemini 2.5 Flash Image on the OneIG-EN benchmark, and Qwen-Image and GPT-4o on LongText-EN. The results on ImgEdit-Bench and multimodal understanding benchmarks are also relatively weaker. In addition, the tables do not clearly highlight the best performance (e.g., missing bold formatting), which makes comparisons less clear.
2. There is no comparison of the total number of model parameters with other methods, which raises questions about the efficiency of the proposed framework.
3. The ablation studies are relatively limited. Important design choices, such as the BSQ dimension, group size, and the weighting factor of the visual loss, are not sufficiently analyzed.
4. The training and inference costs are not reported. In particular, reporting the GPU hours for each training stage would help readers better understand the practical cost of the method. Moreover, although the paper claims that the proposed method can significantly accelerate generation, quantitative evidence supporting this claim is missing.

---

> ### Author Rebuttal · Authors · 2026-03-31
>
> Dear Reviewer VQ3s,
>
> We sincerely thank you for your time and thoughtful, constructive feedback. Below, we provide detailed responses to each weakness (W) and question (Q).
> ***
> >**[W1, Q1] Performance on multiple benchmarks and scaling effects**
>
> Direct comparisons with models such as Qwen-Image, GPT-4o, and Gemini 2.5 Flash Image are not entirely fair for several reasons:
> * **Model scale.** UniAR uses an 8B LLM with a 2.5B DiT, whereas Qwen-Image uses a 7B LLM with a 20B DiT. The sizes of GPT-4o and Gemini are not publicly available, but they are widely believed to be much larger.
> * **Training data scale.** UniAR was likely trained with less data.
> * **Model objective.** Qwen-Image is a generation-only model, whereas UniAR is a unified model.
>
> Among models of similar scale, **UniAR already performs strongly on text rendering and image editing** (see W2).
>
> We explore the scaling effect on reconstruction. Row 1 is the original UniAR; Row 2 scales the decoder to 8B based on Row 1; and Row 3 further increases the BSQ dimension to 128. The results show that **UniAR exhibits well scaling behavior**.
> |Visual Decoder Size|BSQ-dim|rFID↓|PSNR↑|SSIM↑|
> |-|-|-|-|-|
> |2.5B|64|1.4|22.7|0.71|
> |8B|64|0.95|22.7|0.73|
> |8B|128|0.74|25.8|0.81|
>
> As in Tab. 4, our model matches or surpasses pure understanding models on most benchmarks and outperforms most unified models, especially on OCR-related and video tasks. Due to resource constraints, we do not use pure-text data or multimodal RL, both of which could further improve understanding. **We therefore believe there is substantial room for improvement through better data, scaling, and post-training.**
>
> We will revise the table for clarity.
> ***
> **[W2, Q2] Comparison of model size**
> We list the parameter sizes of the major open-source models and compared the generation performance. **UniAR achieves strong performance among models of similar scale**. We will include a more detailed comparison of model sizes in the revised version.
> ||#params|GenEval|OneIG|LongText|
> |-|-|-|-|-|
> |BAGEL|7B+7B|0.88|0.24|0.37|
> |OmniGen2|3B+4B|0.86|0.68|0.56|
> |UniWorldv1|7B+12B|0.84|-|-|
> |UniAR|8B+2.5B|0.88|0.87|0.92|
> ***
> **[W3, Q3] Ablation studies and empirical considerations behind key design choices**
>
> We ablate the BSQ dimension in early experiments with a 1.7B LLM trained on 150B tokens.
> ||avg|ai2d|ocrbench|textvqa|infographics|chartqa|docvqa|mmmu|mmstar|rlwdqa|mmb_en|mmb_cn|
> |-|-|-|-|-|-|-|-|-|-|-|-|-|
> |continuous tokens (baseline)|72.2|76.8|83.9|80.0|71.8|80.5|91.2|43.9|52.5|63.9|76.3|73.7|
> |quantized tokens (BSQ dim = 32)|69.9|74.9|77.6|77.6|66.7|77.60|88.5|44.3|50.4|62.9|76.1|72.3|
> |quantized tokens (BSQ dim = 64)|70.6|75.7|80.1|77.9|66.7|78.8|88.6|44.3|50.8|63.9|75.8|74.1|
>
> As shown in the table, increasing the BSQ dim from 32 to 64 improves performance. Moreover, compared with the continuous-token baseline, the BSQ-quantized tokens reduce overall performance by only about 1.5 points, which is acceptable.
>
> Regarding the weighting factor of the visual loss during joint training, we find that our training scheme is robust to this hyperparameter, ranging from 1.0 to 10.0. Specifically, we adopt square-root-normalized per-token loss [1] during joint training to balance the contributions of text tokens for visual-language understanding and visual tokens for visual generation, making the visual-loss weighting factor less sensitive. Finally, we set the weighting factor of the visual loss to 10.0 in our final experiment.
> ***
> **[W4， Q2] Training and inference costs**
>
> We provide the data scale and GPU hours of each training stage as below:
> |Stage|PT-8K|PT-32K|SFT|RL|
> |-|-|-|-|-|
> |tokens/samples|800B tokens|200B tokens|50B tokens|19.2k samples|
> |GPU hours|~19k|~10k|~2k|~1.9k|
>
> We also provide the inference time. We report the time required to generate a 1024-resolution image. UniAR achieves significantly faster generation, mainly due to its higher 32x downsampling ratio, which quadratically reduces the number of prediction steps. When the decoder uses upsampling, the number of predicted tokens is further reduced to 256, leading to an additional speedup.
> ||downsample ratio|#token|time(s)|
> |-|-|-|-|
> |Januspro(7B)|x16|4096|101.9|
> |XOmni(7B)|x16|4096|119.7|
> |UniAR(8B) w/o decoder upsample|x32|1024|53.5|
> |UniAR(8B) w/ decoder upsample|x64|256|13.0|
>
> We **do not** apply any additional speed optimization and use the standard Transformers lib. As an autoregressive model, UniAR can also benefit from mature LLM acceleration frameworks such as vLLM and SGLang.
>
> [1] Qwen3-VL Technical Report
>
> ***
> **Discussion of limitations.**
> Due to resource constraints, we do not include pure-text data in joint pre-training, leaving considerable room for optimizing the data mixture for both understanding and generation. In addition, our RL training is currently limited to image generation. Moreover, we leave the exploration of data and model scaling to future work. We will incorporate these discussions in the revised version.

---

> > ### Author Rebuttal · Reviewer_VQ3s · 2026-04-03
> >
> > Thank you for the authors’ response. My main concerns have largely been addressed. I have one additional question:
> >
> > Does the 32× downsampling ratio of the visual tokenizer lead to a loss of fine-grained details in image reconstruction or grounding tasks? Could the authors provide a more comprehensive and fair comparison of different tokenizer designs?

---

> > > ### Author Response · Authors · 2026-04-06
> > >
> > > Dear Reviewer,
> > >
> > > We sincerely thank you for your feedback. We are glad that our previous response largely addressed your concerns. For your follow-up questions, we provide the following clarifications.
> > >
> > > ***
> > > > **[Q1] Clarifications on the downsampling ratio**
> > >
> > > We would like to clarify that **the downsampling ratio of the visual tokenizer and that of autoregressive prediction refer to two different concepts**:
> > > * `Visual tokenizer downsampling ratio.`
> > > In UniAR, the downsampling ratio of the visual tokenizer corresponds to a patch size of 16, which is consistent with common ViT-based designs such as SigLIP2 [1] and Qwen3-VL [2]. Therefore, the visual tokenizer **does not** lead to a loss of fine-grained details in image reconstruction or grounding tasks.
> > > * `Auto-regressive downsampling ratio.`
> > > The 32x downsampling ratio mentioned in our previous response actually refers to the downsampling ratio in autoregressive prediction. Specifically, for a 1024 resolution image, UniAR only requires $\frac{1024}{32}\times\frac{1024}{32}$ next-token prediction steps. More concretely, at each step, UniAR adopts the **parallel bitwise prediction** mechanism described in Sec. 3.2 of the main paper to concurrently predict the BSQ indices for a $2\times 2$ spatial region.
> > >
> > > In summary, the tokenizer in UniAR operates at a 16x downsampling ratio and therefore does not compromise fine-grained visual information. The 32x ratio comes from the autoregressive prediction process, where the number of prediction steps is reduced through the parallel bitwise prediction mechanism, substantially improving inference efficiency. We will clarify this point in the revised version.
> > >
> > > ***
> > > > **[Q2] Discussions on different tokenizer designs**
> > >
> > > Thank you for this helpful suggestion. As clarified in Q1, the tokenizer in UniAR uses a 16x downsampling ratio, which is consistent with common tokenizer designs. We choose X-Omni [3] as the baseline because it is the most comparable prior design: it also adopts 16x tokenizer downsampling and is trained with understanding-oriented objectives without any reconstruction loss.
> > >
> > > The key differences between X-Omni and UniAR are:
> > > * Quantization method: X-Omni adopts conventional vector quantization, which limits the vocabulary size, whereas UniAR uses binary spherical quantization, enabling a much larger vocabulary while keeping the overhead manageable.
> > > * Multi-level feature fusion: X-Omni uses only the last-layer features, while UniAR fuses multi-granularity features from different layers, allowing it to capture both high-level semantics and low-level details.
> > >
> > > These two differences are exactly the core innovations of the UniAR tokenizer design. To verify their effectiveness, we evaluate the reconstruction performance of different tokenizers on ImageNet. All evaluations in the table below are conducted at 1024 resolution to ensure a fair comparison. Notably, the X-Omni tokenizer has been adopted by GLM-Image [4], indicating it to be a strong baseline. We observe that, despite using smaller encoder and decoder models than X-Omni, UniAR achieves substantially better reconstruction performance. This result confirms the effectiveness of the key design choices in UniAR, namely binary quantization and multi-level feature fusion. In addition, we provide reconstruction visualizations below for qualitative analysis (please see [this figure](https://anonymous.4open.science/r/rebuttal-figures-E33D/xomni-uniar-compare.jpg)).
> > >
> > > |method|quantize method|multi-level feature|downsampling ratio|reconstruction loss|bsq dim|encoder size|decoder size|rFID↓|PSNR↑|SSIM↑|
> > > |---|---|:---:|:---:|:---:|---|---|---|---:|---:|---:|
> > > |X-Omni|VQ|$\times$|16x|$\times$|-|1B|12B|4.87|17.1|0.45|
> > > |UniAR|BSQ|$\checkmark$|16x|$\times$|64|0.4B|2.5B|1.40|22.7|0.71|
> > > |UniAR*|BSQ|$\checkmark$|16x|$\times$|64|0.4B|8B|0.95|22.7|0.73|
> > > |UniAR**|BSQ|$\checkmark$|16x|$\times$|128|0.4B|8B|0.74|25.8|0.81|
> > >
> > > ***
> > >
> > > We sincerely thank you again for your time and constructive suggestions, which are very helpful for improving our paper. We will incorporate these discussions into the revised version. We hope our response has clarified your questions.
> > >
> > > Best regards,
> > > The authors
> > >
> > > ***
> > > [1] SigLIP 2: Multilingual Vision-Language Encoders with Improved Semantic Understanding, Localization, and Dense Features
> > > [2] Qwen3-VL Technical Report
> > > [3] X-Omni: Reinforcement Learning Makes Discrete Autoregressive Image Generative Models Great Again
> > > [4] GLM-Image: Auto-regressive for Dense-knowledge and High-fidelity Image Generation

---

### Official Review · Reviewer_kMkR · 2026-03-13

**Soundness:** 3
**Presentation:** 3
**Significance:** 2
**Originality:** 2
**Overall Recommendation:** 4
**Confidence:** 4

**Summary:**

The paper proposes UNIAR, an autoregressive model that predicts text and visual tokens jointly. The design includes bitwise visual tokenization, multi-level visual features, and a parallel bitwise prediction mechanism to reduce sequence length. Experiments show strong results on image generation benchmarks and competitive performance on multimodal understanding tasks.

**Compliance With Llm Reviewing Policy:**

Affirmed.

**Final Justification:**

I appreciate the efforts in the rebuttal. My concerns have been addressed, and I keep my score as weak accept.

**Key Questions For Authors:**

Please refer to the weakness.

**Limitations:**

yes

**Strengths And Weaknesses:**

Strengths:

1. The motivation is clear. Using a single tokenizer to unify understanding and generation is a reasonable direction and addresses a real limitation in previous dual-tokenizer systems.

2. The bitwise quantization design is interesting and allows a very large implicit visual vocabulary with low overhead.

3. Experimental results on generation benchmarks (e.g., GenEval) appear strong and competitive with recent unified models.

Weaknes:

1. The paper mainly emphasizes generation performance; the improvements on multimodal understanding benchmarks seem more modest.

2. It is not entirely clear how much of the gain comes from the tokenizer design versus scale of training.

3. The efficiency claims would benefit from more detailed runtime or inference cost analysis.

---

> ### Author Rebuttal · Authors · 2026-03-31
>
> Dear Reviewer kMkR,
>
> We sincerely thank you for your time and thoughtful, constructive feedback. Below, we provide detailed responses to each weakness (W).
> ***
> > **[W1] Performance on multimodal benchmarks**
>
> Thank you for pointing this out. While our work primarily focuses on generation, we also aim to maintain strong multimodal understanding capability. As shown in Tab. 4 of our paper, our model achieves comparable or even superior performance on most benchmarks relative to pure understanding models, and surpasses most unified models, particularly on OCR-related and video benchmarks.
>
> It is worth noting that, due to resource constraints, we do not incorporate pure-text data during training, nor do we apply multimodal understanding RL in post-training — both of which have been shown to be effective for enhancing multimodal comprehension. We believe our model has significant potential for further improvement on understanding tasks with a more balanced data mixture, higher-quality data, and multimodal RL.
> ***
> > **[W2] The source of the gains: tokenizer design or training scale**
>
> To disentangle the effect of tokenizer design from training scale, we conduct controlled studies on two key designs of our tokenizer: (1) bitwise quantization, which enables a vastly large codebook, and (2) multi-level feature fusion, which improves feature representation.
>
> `Bitwise quantization` We first evaluate BSQ on a 1.7B LLM trained on 150B tokens of curated core set. As shown below, BSQ quantization achieves performance close to the continuous-token baseline, with only a modest drop. Moreover, increasing the BSQ dimension from 32 to 64 further improves the performance of the quantized tokenizer. In addition, quantization also improves training efficiency; please refer to our response to W3 for detailed runtime results.
>
> ||avg|ai2d|ocrbench|textvqa|infographics|chartqa|docvqa|mmmu|mmstar|rlwdqa|mmb_en|mmb_cn|
> |---|---|---|---|---|---|---|---|---|---|---|---|---|
> |continuous tokens (baseline)|72.2|76.8|83.9|80.0|71.8|80.5|91.2|43.9|52.5|63.9|76.3|73.7|
> |quantized tokens (BSQ dim = 32)|69.9|74.9|77.6|77.6|66.7|77.60|88.5|44.3|50.4|62.9|76.1|72.3|
> |quantized tokens (BSQ dim = 64)|70.6|75.7|80.1|77.9|66.7|78.8|88.6|44.3|50.8|63.9|75.8|74.1|
>
> `Multi-level feature fusion`
> Figure 3 in the main paper illustrates the importance of multi-level features for image reconstruction: deep-layer features mainly capture high-level semantics but lose high-frequency details, whereas shallow-layer features better preserve fine-grained information. Combining features from multiple levels yields the best reconstruction quality.
>
> We thank the reviewer for raising this important question. The results suggest that the gains are mainly attributable to the tokenizer design, rather than to training scale alone. We will add this discussion to our revised version.
>
> ***
> > **[W3] More detailed runtime and inference cost analysis**
>
> Regarding training efficiency, discrete visual tokens improve pre-training throughput by approximately 30% over continuous tokens at a sequence length of 8K, as visual inputs can be pre-tokenized and stored as bit-packed representations offline. Detailed training elapsed times are provided below.
> ||elapsed time per iteration|
> |-|-|
> |continuous visual tokens|35.4s|
> |discrete visual tokens|24.5s (-30.8%)|
>
> Regarding the inference efficiency for image generation, we compare UniAR with Janus-Pro and X-Omni. All models are evaluated on the same A100 GPU without CFG, and we report the time required to generate a 1024-resolution image. UniAR achieves significantly faster generation, mainly due to its higher 32x downsampling ratio, which quadratically reduces the number of prediction steps. When the decoder uses upsampling, the number of predicted tokens is further reduced to 256, leading to an additional speedup.
>
> We do not apply any additional speed optimization and use the standard Transformers library. Since UniAR follows autoregressive paradigm, it can readily benefit from mature LLM acceleration frameworks such as vLLM and SGLang, which could further improve its inference speed.
> ||downsample ratio|#token|time(s)|
> |-|-|-|-|
> |Janus-pro(7B)|x16|4096|101.9|
> |X-Omni(7B)|x16|4096|119.7|
> |UniAR(8B) w/o decoder upsample|x32|1024|53.5|
> |UniAR(8B) w/ decoder upsample|x64|256|13.0|

---

> > ### Author Rebuttal · Reviewer_kMkR · 2026-04-02
> >
> > Thank you for the detailed responses. The additional analyses and clarifications regarding multimodal performance, tokenizer design, and efficiency are helpful and address my previous concerns.
> >
> > I appreciate the controlled studies disentangling tokenizer design from training scale, as well as the added discussion on efficiency. These revisions improve the clarity and completeness of the paper.
> >
> > I will keep my original score.

---

> > > ### Author Response · Authors · 2026-04-05
> > >
> > > Dear Reviewer,
> > >
> > > We are glad that our previous response helped address your questions, and we are encouraged that you found our motivation clear, our design interesting, and our results "strong and competitive." We sincerely thank you again for your constructive feedback, and we will incorporate these points into the revised version.
> > >
> > > Best regards,
> > > The authors

---

### Decision · Program_Chairs · 2026-04-30

**Decision:**

Accept (regular)

**Comment:**

The submission addresses the problem of unifying understanding and generation, tackled through shared discrete tokenization. All reviewers agreed with the merit of the paper, and although they stopped short of finding it exciting, they agreed that it has enough merit to appear in ICML. The AC agrees and recommends acceptance. @the authors, please carefully take into account all reviewers' comments and exchanges, and address them in the camera-ready for greater impact by the paper.